

# Fusion of text and graph information for machine learning problems on networks

Ilya Makarov[1,2], Mikhail Makarov[1] and Dmitrii Kiselev[1]

[1] HSE University, Moscow, Russia
[2] University of Ljubljana, Ljubljana, Slovenia

## ABSTRACT

Today, increased attention is drawn towards network representation learning, a technique that maps nodes of a network into vectors of a low-dimensional embedding space. A network embedding constructed this way aims to preserve nodes similarity and other specific network properties. Embedding vectors can later be used for downstream machine learning problems, such as node classification, link prediction and network visualization. Naturally, some networks have text information associated with them. For instance, in a citation network, each node is a scientific paper associated with its abstract or title; in a social network, all users may be viewed as nodes of a network and posts of each user as textual attributes. In this work, we explore how combining existing methods of text and network embeddings can increase accuracy for downstream tasks and propose modifications to popular architectures to better capture textual information in network embedding and fusion frameworks.

## INTRODUCTION

Many real-world data can be modeled as graphs: citation networks, social networks, knowledge databases. Ability to analyze such data structures is crucial for a great variety of applications. For instance, when social networks try to get new users to subscribe, they need to solve a *link prediction problem* (LPP) (*Backstrom & Leskovec, 2011*). Telecom companies' marketing departments might want to segment users according to their behavior within a network of calls, which can be stated as *a node clustering* problem (*Zhu et al., 2011*). Biologists need to find out the structural roles of proteins via analyzing their interaction network, requiring a solution for *node classification* problem (*Do, Le & Le, 2020*). *Recommendations* of future collaborations can be constructed via combination of graph topology and researcher feature engineering (*Makarov, Bulanov & Zhukov, 2017*; *Makarov et al., 2018*; *Makarov & Gerasimova, 2019a*, *2019b*).

All problems mentioned above correspond to classic machine learning problems applied to networks, with every network represented by a graph and attributes of its components, such as nodes or edges. Solving machine learning problems on network data require vector representation for object features, including graph structure. To be able to solve these problems, one has to develop the efficient representation of a network that

Corresponding authors
Ilya Makarov, iamakarov@hse.ru
Dmitrii Kiselev, dkiseljov@hse.ru

will preserve attribute features and graph structure, and will be feasible for existing machine learning frameworks.

Historically, the first way to represent a graph is the adjacency matrix. This representation has two significant drawbacks. Firstly, it captures only direct relationships between nodes. Secondly, for real-world networks, the adjacency matrix tends to be very sparse and does not directly represent structural features apart from first-order proximity.

Network Representation Learning (NRL) techniques were created to mitigate the problems mentioned above. The main idea of NRL is to map nodes (or edges) of a network into low-dimensional space preserving their topological structure from the network. The first NRL methods were mostly based on matrix factorization (*Roweis & Saul, 2000*; *Belkin & Niyogi, 2002*). These methods do solve the dimensionality problem but are highly computationally expensive. A more advanced approaches use random walks on networks to approximate different kinds of similarity matrices (*Perozzi, Al-Rfou & Skiena, 2014*; *Grover & Leskovec, 2016*). These methods are very scalable and, therefore, can be applied even to large networks.

Quite often, nodes of a network have different kinds of attributes associated with them. This work is concerned with one type of attributes—textual information. The problem of efficient representation of textual information is very similar to the same problem with graphs. The classic techniques, such as Bag of Words (BoW) suggested by *Harris (1954)* and Term Frequency-Inverse Document Frequency (TF-IDF) suggested by *Salton & Buckley (1988)*, encode each word as a one-hot vector and represent a document as a sum of representations of all words (using certain coefficients). These methods are straightforward but produce very sparse representations and do not consider the order of words. A more advanced approach, called Word2Vec (*Mikolov et al., 2013*), employs a Skip-Gram model to learn semantics of words through their context. This method produces dense low-dimensional embeddings, thus gaining an advantage over the classic approaches. There are some extensions of Word2Vec like *Pagliardini, Gupta & Jaggi (2017)* and *Mikolov & Le (2014)*. Their aim is to learn document embeddings instead of embeddings for separate words. The most advanced models use bidirectional transformers (*Reimers & Gurevych, 2019*) to learn sensible embeddings.

The fusion of graph and text information for representation learning is still an area that is not well researched. The most straightforward approach is to learn network and text embeddings separately and then concatenate them to produce the final embedding. More sophisticated approaches include TADW (*Yang et al., 2015*), which incorporates text attributes into a matrix factorization problem. TriDNR (*Pan et al., 2016*) uses combined loss between Doc2Vec and DeepWalk algorithms. Finally, GCN (*Kipf & Welling, 2016*) and its variations use graph neural networks to take node attributes into account.

In this work, the following contributions are made:

1. Different combinations of network and text embeddings are studied to improve the downstream tasks quality.

2. Some modifications are proposed to existing architectures to better take into account text and graph information and the way how they are fused.

3. Comprehensive comparison of existing methods is performed on node classification, link prediction and visualization problems.

The paper is structured as follows. We start with a brief explanation of related work and the choice of models. Then, we describe experiment methodology: used datasets, training and validation schemes for all the models and machine learning problems on networks. Next, we explain the obtained results. Finally, we provide ideas for further enhancement of fusion techniques in discussion and conclude our study. All sections describe the content in the following order: text embeddings, then structural network embeddings, and finally, fusion models of text and network data.

## RELATED WORK

In the real-life scenario, networks are often accompanied by additional information. In this work, the main focus is on one particular case, where each node of a network is associated with text information. Below, we shortly discuss the chosen text and network embedding models, as well as several popular strategies of information fusion for the considered problem.

### Text embeddings

#### Latent dirichlet allocation (LDA)

*Martnez & Kak (2001)* propose the topic modeling techniques. It is a Bayesian generative probabilistic model for document clustering. Each document embedding is a vector of weights for underlying topics, where topics consist of several words with individual weights.

#### Word2Vec

The idea of *Mikolov et al. (2013)* is to predict context from a word (Skip-gram) or a word from its context (Continuous Bag of Word or just, CBoW).

#### Sent2Vec

It is an extension of Word2Vec CBoW model, which was explicitly designed to improve sentence embeddings (*Pagliardini, Gupta & Jaggi, 2017*). Firstly, it also learns embeddings for word n-grams. Secondly, it uses a whole sentence as a context window. Such an approach allows receiving better sentence embedding with n-gram aggregations.

#### Doc2Vec

*Mikolov & Le (2014)* extend Word2Vec approach even further to learn continuous representations for texts of variable length (starting from a short phrase to very long articles). Its main distinction from Sent2Vec is that Doc2Vec can preserve text context for very long sequences of words. Doc2Vec additionally creates a lookup table with text embeddings. When a target word is predicted, this vector is concatenated to a source word vector.

### SBERT

SBERT (*Reimers & Gurevych, 2019*) is an extension of classic BERT (*Devlin et al., 2018*). The main difference is that SBERT is trained in contrastive fashion using Siamese architecture. In comparison to bidirectional autoencoder with a self-attention mechanism, it uses more advanced pooling strategies.

### Ernie

*Sun et al. (2020)* suggest increasing the number of pretraining objectives to capture corpora's lexical, syntactic, and semantic information. The framework uses continual multi-task learning to sequentially learn new tasks without "forgetting" the previous ones.

## Network embeddings

There is a large variety of network embedding models for different cases. In the current work, we use in experiments only three models without node attributes, typically called structural embeddings because of their nature to learn graph structure independently of node attributes.

### DeepWalk

Invented by *Perozzi, Al-Rfou & Skiena (2014)*, the model samples random walks and learns embeddings using a skip-gram approach similar to *Mikolov et al. (2013)*.

### Node2Vec

*Grover & Leskovec (2016)* propose a more efficient realization of the random walk idea. It balances between breadth-first and depth-first searches to keep tradeoff between local and global graph structures.

### HOPE

*Ou et al. (2016)* employ matrix factorization technique to directly reconstruct asymmetric distance measures like Katz index, Adamic-Adar or common neighbors. So it preserves asymmetric transitivity, which is important property of directed graphs.

## Naive mixture

The most straightforward method to fuse graph and text information is to learn graph and text embeddings independently. Then combine two types of embeddings, concatenating them. This method has the following advantages:

1. Graph and text embeddings have been researched separately for quite a long time, so there are plenty of available methods/libraries etc.
2. Because embeddings for nodes and texts are learned individually, they provide a lot of freedom to choose a different dimension for graph and text embeddings, pre-train text embeddings on an entirely different corpus.

The main disadvantage is evident: text information is not taken into account while learning graph embedding and vice versa. It is essential, because two nodes might have the same distance in graph proximity but completely different semantic meaning.

### Advanced mixture

Below, we describe chosen well-known fusion methods to use for our comparison of information fusion methods.

#### Text attributed deep walk (TADW)

One of the first attempts to incorporate text information into network representation learning was made in the TADW algorithm (*Yang et al. (2015)*). The main idea was to enrich ordinary DeepWalk algorithm by taking into account text attributes. The authors prove that DeepWalk performs a matrix factorization process and extend it with TF-IDF feature matrix.

#### Tri-Party Deep Network Representation (TriDNR)

*Pan et al. (2016)* try to solve two issues of TADW: computational complexity of matrix factorization and missed word order in TF-IDF matrix encoding of texts. As the name suggests, the algorithm learns the network representation using three sources: graph, text and label information. DeepWalk algorithm is applied to capture graph information. For text and label information, refined Doc2Vec is used.

#### Graph Convolutional Network (GCN)

*Kipf & Welling (2016)* propose Graph Convolution Networks (GCN) as a light-weight approximation for the spectral convolution. This method provides better computational efficiency for semi-supervised tasks, such as link prediction or node classification. One of the main advantages of GCNs is their ability to account for node attributes. GCN works similarly to the fully-connected layers for neural networks. It multiplies weight matrices with the original features but masking them with an adjacency matrix. Such a method allows to account only for node neighbors and node representation from the previous layer.

#### Graph attention networks (GAT)

*Veličković et al. (2017)* utilize the idea of the self-attention mechanism of *Vaswani et al. (2017)* for network data. Such an approach allows to balance the weights of neighbors in node embedding according to structure and node attributes. Because masked self-attention does not require knowing the graph structure upfront, this model could be used inductively.

#### Graph SAmple and aggreGatE (GraphSAGE)

*Hamilton, Ying & Leskovec (2017)* suggest using sampling over node neighborhood to learn final embedding. It provides more scalability and different choices for learnable aggregation functions.

#### Graph InfoClust (GIC)

GIC (*Mavromatis & Karypis, 2020*) leverages the cluster-level information to any graph neural network (GNN) encoder. They propose to add a new part to the loss maximizing mutual information between node representations on both cluster and global levels.

Network substructures such as clusters usually correlate with node labels, and link creation inside a cluster is more likely by their definition.

Generally, described fusion methods outperform text or network embeddings. Still, there is some room for improvement: as for now, most researchers use BoW or TF-IDF to produce input feature matrix for fusion methods, such as TADW and GCN. It is promising to see how the combination of advanced text embedding techniques with these methods might improve the accuracy of machine learning tasks. Also, one might be interested in enhancing GCN architecture by adding simultaneously trainable word embeddings to the network.

## EXPERIMENTS

This section explains the experiment pipeline to determine whether the fusion of text and graph information helps improve the quality of the downstream tasks. Firstly, we describe the choice of datasets. Then, we define the process of constructing text embeddings after text preprocessing. Next, we describe our choice of network embeddings and their hyperparameters. Finally, the fusion techniques and hyperparameters are provided. In the end, we describe the training and validation scheme for node classification and link prediction tasks.

### Datasets

To be able to compare different kinds of algorithms described above, the chosen dataset should possess the following properties:

1. It should have a graph structure, i.e. it should contain entities and relations between them.
2. At least some of the nodes should have text associated with it. It is important to note that texts associated with nodes should be in raw format (e.g., not in embedding format already, such as BoW). Although it is not required for every node to have text associated with it, the more nodes have it, the better the quality is.
3. At least some nodes should be associated with labels. This property is necessary to state the node classification problem.

Below, we describe three main datasets chosen as benchmarks for network-related machine learning problems and satisfying conditions above.

***Cora*** (*Sen et al., 2008*). Cora dataset is a citation network, in which each node represents a scientific paper, and each link shows that one article cites another one. There are 2708 nodes and 5429 edges in the network. Each node has text with a short description (abstract) of the paper. Average text length in words is 130. All nodes are grouped into seven classes: Neural Networks, Rule Learning, Reinforcement Learning, Probabilistic Methods, Theory, Case-Based, Genetic Algorithms. The network does not contain any isolated nodes.

***CiteSeer-M10*** (*Lim & Buntine, 2016*). This dataset is a subset of original CiteSeer data, which contains scientific publications in different disciplines grouped into ten different classes. M10 version consists of 38,996 nodes and 76,630 edges. However, only 10,310

nodes have the text (paper title) and label information associated with them. Average text length in words is 9. In this case, text information contains only name of the paper (rather than the abstract). Some of the nodes are isolated, which makes this dataset generally more problematic than the previous one.

**DBLP** (*Yang & Leskovec, 2015*). DBLP is a bibliographic system for computer science publications. Citations might connect publications (described by the title). In this work, we follow the comparison setting suggested by *Pan et al. (2016)*, and consider only subset of the network, containing 60,744 nodes (all accompanied with text and label attributes) and 52,890 edges. Average text length in words is 8.

## Text embeddings

The first part of the experiments is mostly concerned with estimating whether textual information alone can efficiently solve machine learning problems on text networks. Intuitively, in case of citation networks, text description should be correlated with the target class (the topic of research), so results on text data can provide a good baseline, using which other types of embeddings are compared. Questions to be addressed in this section:

1. Whether advanced text embedding techniques (Sent2Vec, Doc2Vec, SBERT) generally outperform classic approaches (such as BoW, TF-IDF) in case of citation networks?
2. How does a share of train data (compared with test) affect the model prediction power?
3. How does average text length influence model quality?
4. Whether models pre-trained on a vast amount of data perform better than models trained "from scratch"?

One of the most crucial steps to start with the problem is text preprocessing. We perform preprocessing before embedding algorithm is applied. We follow the standard pipeline.

Firstly, we remove all special symbols and switch the case to lower. Next, we remove stop words. Stop words are the set of most frequently used words in a language like "also". In addition, we filter the most frequent words for the current dataset (appear in more than 70% of texts) and the rarest (appear less than three times). Finally, each token is converted to the corresponding lemma, which is the form of a word presented in the dictionary.

Bag of Words and TF-IDF models use only unigrams as input since datasets are relatively small, and choosing higher ngram_range will lead to poor generalization. For LDA we use Gensim implementation (https://radimrehurek.com/gensim/models/ldamodel.html) with following hyperparameters: number of topics (efficient embedding size) = 20, $\alpha = 0.1$, $\beta = 0.1$.

Word2Vec, Doc2Vec and Sent2Vec models were used with and without pretraining. Trained models are based on English Wikipedia. Local training of Word2vec and Doc2vec was performed using Gensim with following hyperparameters: window size is equal to 5, $\alpha = 0.025$, `ns_exponent` `parameter` equals to 0.75.

SBERT was pre-trained (https://github.com/UKPLab/sentence-transformers) on SNLI dataset *Bowman et al. (2015)*, which consists of 570,000 sentence pairs divided into 3

classes: contradiction, entailment and neutral. We use the original pre-trained version of ERNIE by Baidu (https://huggingface.co/nghuyong/ernie-2.0-en). To achieve sentence embedding from Ernie, we average last hidden state for all its tokens.

## Network embeddings

In citation networks, papers from one field tend to cite each other more frequently than articles from other areas. Therefore, the graph structure should give significant insights into the node classification and link prediction tasks. Another critical issue is comparing how well network embeddings perform compared to text embeddings on different datasets.

Three network embedding methods were selected for the experiments with structural network embeddings: HOPE, Node2Vec and DeepWalk. The reason for such a choice is quite straightforward: these methods tend to outperform others in most settings of structural network embeddings (*Makarov et al. (2021)*). We use GEM implementation of HOPE (https://github.com/palash1992/GEM). Hyperparameter $\beta$ is chosen to be 0.01 (as used in other papers). For DeepWalk original implementation (https://github.com/phanein/deepwalk) is used, with the following hyperparameters: vector size—10, number of walks per-vertex—80, window size—10. Node2Vec also follows original implementation (https://github.com/aditya-grover/node2vec) with the following hyperparameters: vector size—10, number of walks per-vertex—80, window size—10 (the same as DeepWalk).

## Fusion of text and graph information
### Naive combination

Text and network embeddings are learned separately. For every node, the final embedding is represented as concatenation of the corresponding text embedding and network embedding. This method can be viewed as a good baseline for fusion methods. In this combination, we use DeepWalk as network embedding similarly to the more comprehensive TADW method. We concatenate it with BoW, also following the approach of TADW. Additionally, we test it with concatenations of Sent2Vec embedding as an advance text encoding approach.

### TADW

Two versions of TADW were constructed with the help of TF-IDF or Sent2Vec for the feature generation. The following hyperparameters are used: vector size = 160, number of iterations = 20, $\lambda$ = 0.2. SVD is used on input feature matrix to reduce its dimension to 200 (as in the original paper).

### TriDNR

All three sources are used: texts, network and labels to get the final embeddings. Only labels from the train set are present, while others are masked. The following hyperparameters are used: vector size = 160 (to match TADW), text weight = 0.8, passes = 50.

### Graph neural networks (GCN, GAT, GraphSAGE and GIC)

In most papers, authors use simple BoW or TF-IDF matrices as a feature matrix for GCN. It might be sensible to experiment with more advanced text embedding techniques to

improve the results, as we have already seen that Sent2Vec or Word2Vec outperform BoW and TF-IDF for some settings. The model is trained for 200 epochs using Adam optimizer. The best model (according to validation results) is saved. The vector size is equal to 64, and the model contains two convolutional layers.

Also, it is interesting to try new modification for GCN architecture. Instead of using a fixed feature matrix as input, one can replace it with a lookup table with learnable embeddings. This way model can simultaneously learn text embeddings as well as network embeddings. In this case, padded sentences of tokens are fed as input, then the lookup table with embeddings is used. Next, to obtain embeddings for sentences, mean and max functions are applied for word embeddings of each sentence. In the end, the rest of the network is treated the same way as for ordinary GCN.

## Training and validation

Input network data consists of nodes of the graph with associated text information and edges between them. Before the validation procedure, text information is preprocessed using the steps described in Section 3.2.

### Node classification

We start with encoding nodes using one of the embedding techniques: text, graph or their fusion. Next, we split the dataset into train and test subsets in different proportions (5%, 10%, 30% and 50% of labeled nodes). Finally, Logistic Regression classifier is trained.

We use Logistic Regression for two reasons. Firstly, almost all learned embeddings are non-linear in nature (except for TF-IDF and BoW). So Logistic Regression is sufficient for the final classification task. Secondly, it could be pipelined in GNN models by simply adding one fully-connected output layer. Such a technique allows training neural networks in an end-to-end fashion.

### Link prediction

The edges of the graph are split randomly into train and test sets with specified train ratios (5%, 10%, 30% and 50% train edges). Then test edges are masked (effectively removed) from the graph.

Usually, most graphs are sparse, so the number of existing edges dramatically less than the number of all possible links. We keep the LPP as a binary classification problem. So in general, LPP has highly imbalanced classes. One of the popular techniques to handle it is to use undersampling of the dominant class. To make the final classifier more powerful, we sample non-existing links between most similar nodes because it is the most probable connection to appear. Existing edges are marked as "ones" and non-existing as "zeros". The same is done for the test set.

The masked graph is then used to learn node embeddings (using text or graph information or both). We use simple element-wise (Hadamard) product of node embeddings as encoding for the corresponding edge, leaving other edge encoder operators for future work (see *Makarov et al., 2019*, *2018a*, *2018b*). Finally, we train Logistic Regression on obtained vectors to classify the presence or absence of the links between pairs of nodes in the graph.

**Table 1 Text methods on Cora for node classification (micro-F1, metric lies between (0,1) and higher value means better results).**

| % Labels | 5% | 10% | 30% | 50% |
|---|---|---|---|---|
| BoW | 0.63 ± 0.01 | 0.68 ± 0.01 | **0.76 ± 0.01** | **0.78 ± 0.01** |
| TF-IDF | 0.35 ± 0.01 | 0.49 ± 0.01 | 0.70 ± 0.01 | 0.76 ± 0.01 |
| LDA | 0.49 ± 0.01 | 0.57 ± 0.01 | 0.60 ± 0.01 | 0.61 ± 0.01 |
| SBERT pretrained | 0.57 ± 0.01 | 0.61 ± 0.01 | 0.68 ± 0.01 | 0.70 ± 0.01 |
| Word2Vec pretrained | 0.34 ± 0.01 | 0.44 ± 0.01 | 0.59 ± 0.01 | 0.63 ± 0.01 |
| Word2Vec (d = 300) | 0.64 ± 0.01 | 0.68 ± 0.01 | 0.70 ± 0.01 | 0.71 ± 0.01 |
| Word2Vec (d = 64) | 0.65 ± 0.01 | 0.68 ± 0.01 | 0.70 ± 0.01 | 0.72 ± 0.01 |
| Doc2Vec pretrained | 0.54 ± 0.01 | 0.61 ± 0.00 | 0.65 ± 0.01 | 0.67 ± 0.01 |
| Doc2Vec (d = 300) | 0.49 ± 0.01 | 0.58 ± 0.01 | 0.66 ± 0.01 | 0.68 ± 0.01 |
| Doc2Vec (d = 64) | 0.50 ± 0.02 | 0.58 ± 0.01 | 0.65 ± 0.00 | 0.67 ± 0.01 |
| Sent2Vec pretrained | 0.63 ± 0.02 | 0.69 ± 0.01 | 0.74 ± 0.01 | 0.77 ± 0.01 |
| Sent2Vec (d = 600) | **0.68 ± 0.02** | **0.72 ± 0.01** | 0.75 ± 0.01 | 0.77 ± 0.01 |
| Sent2Vec (d = 64) | **0.68 ± 0.02** | **0.72 ± 0.01** | 0.75 ± 0.01 | 0.77 ± 0.01 |
| Ernie pretrained | 0.43 ± 0.01 | 0.52 ± 0.01 | 0.62 ± 0.01 | 0.65 ± 0.01 |

**Note:**
The best values with respect to confidence intervals are highlighted in bold.

### Evaluation process

The procedures described above are repeated five times with random train/test splits for different train/test ratio values in both cases. The mean and standard deviation of the results are reported for $F_1$ quality metric.

For Logistic Regression sklearn (https://scikit-learn.org/stable/modules/generated/sklearn.linear_model.LogisticRegression.html) implementation (in python) is used with lbfgs solver, $L_2$ penalty and $C = 1$. For multi-class classification (number of classes greater than 2) One-Vs-Rest setting is applied, which means that one classifier is trained for every class, besides each model use samples from all other classes as "zero" class.

## RESULTS

Firstly, we present the node classification task results, then we discuss results for link prediction and explain node visualization. Similarly to other blocks, subsections have the following order: text embedding, network embedding, their fusion.

### Node classification

#### Text methods

Table 1 shows the comparison between text approaches on Cora dataset. Classic techniques show very promising metrics, especially when the percentage of labeled nodes is not very small. The best algorithm is the Bag of Words, which outperforms every other classic method. It also shows quite a good quality for different percentage of known labels. TF-IDF performs similarly on 30% and 50% of labeled nodes but degrades significantly on the lower values. Although LDA results are not very high, it shows consistent results across different shares of labeled nodes.

**Table 2 Text methods on Citeseer-M10 for node classification (micro-F1, metric lies between (0,1) and higher value means better results).**

| % Labels | 5% | 10% | 30% | 50% |
|---|---|---|---|---|
| BoW | 0.62 ± 0.00 | 0.66 ± 0.00 | 0.73 ± 0.01 | **0.76 ± 0.01** |
| TF-IDF | 0.61 ± 0.01 | 0.66 ± 0.01 | 0.72 ± 0.01 | 0.75 ± 0.00 |
| LDA | 0.37 ± 0.01 | 0.38 ± 0.00 | 0.39 ± 0.00 | 0.39 ± 0.00 |
| SBERT pretrained | 0.66 ± 0.00 | 0.68 ± 0.00 | 0.72 ± 0.01 | 0.73 ± 0.01 |
| Word2Vec pretrained | 0.67 ± 0.00 | 0.69 ± 0.00 | 0.72 ± 0.00 | 0.73 ± 0.01 |
| Word2Vec (d = 300) | 0.55 ± 0.00 | 0.57 ± 0.00 | 0.59 ± 0.00 | 0.60 ± 0.01 |
| Word2Vec (d = 64) | 0.58 ± 0.00 | 0.59 ± 0.00 | 0.61 ± 0.00 | 0.62 ± 0.01 |
| Doc2Vec pretrained | **0.68 ± 0.00** | **0.70 ± 0.00** | **0.74 ± 0.01** | 0.75 ± 0.00 |
| Doc2Vec (d = 300) | 0.53 ± 0.00 | 0.56 ± 0.00 | 0.59 ± 0.00 | 0.61 ± 0.00 |
| Doc2Vec (d = 64) | 0.56 ± 0.01 | 0.59 ± 0.00 | 0.62 ± 0.00 | 0.63 ± 0.00 |
| Sent2Vec pretrained | **0.68 ± 0.00** | **0.70 ± 0.00** | 0.73 ± 0.01 | 0.75 ± 0.01 |
| Sent2Vec (d = 600) | 0.64 ± 0.01 | 0.66 ± 0.00 | 0.70 ± 0.01 | 0.71 ± 0.01 |
| Sent2Vec (d = 64) | 0.63 ± 0.01 | 0.65 ± 0.00 | 0.68 ± 0.00 | 0.69 ± 0.01 |
| Ernie pretrained | 0.59 ± 0.01 | 0.63 ± 0.00 | 0.67 ± 0.00 | 0.68 ± 0.01 |

Note:
The best values with respect to confidence intervals are highlighted in bold.

Despite being significantly simpler and easier to train, advanced models trained from scratch generally performed better than their pre-trained counterparts. That is probably because language, which is used to describe the scientific paper in computer science, significantly differs from the language used in news or (average) Wikipedia article. The best model out of advanced text embeddings is Sent2Vec (trained from scratch). It also shows consistent results concerning the different share of labeled nodes. Another insight is that advanced embeddings could not beat the Bag of Words technique when the labeled data share is high enough. The nature of the data might explain this. Abstract for scientific papers is just a set of keywords. In this case, the Bag of Words hypothesis is applied very well. However, when there is a small percentage of labeled data (which is the practical case), advanced embeddings significantly outperform Bag of Words, which means that they tend to generalize better. One can note that choosing different values for embeddings dimension does not influence the results significantly.

Table 2 present the results on CiteSeer-M10 dataset. This dataset differs from the first one because texts are significantly shorter, but the total amount of nodes is bigger. One can note that although Bag of Words is still the best out of the classic techniques, TF-IDF performs almost as good on CiteSeer-M10. The quality does not degrade significantly when the percentage of labeled nodes becomes smaller. The reason for that is two-fold: firstly, the attributed network is large, so it is easier to generalize even with a smaller number of labels. Secondly, text length is much smaller, therefore, there is less "variation" in the data. Similarly to Cora, LDA performed even poorer than on the first dataset because it is much harder to extract "topics" from a few words for small datasets.

Considering advanced text embedding techniques, all (except for Ernie) architectures (at least in one of the configurations) outperform classic methods when the percentage of

**Table 3 Text methods on DBLP for node classification (micro-F1, metric lies between (0,1) and higher value means better results).**

| % Labels | 5% | 11% | 30% | 50% |
|---|---|---|---|---|
| BoW | 0.75 ± 0.00 | 0.77 ± 0.00 | **0.79 ± 0.00** | **0.80 ± 0.00** |
| TF-IDF | 0.74 ± 0.01 | 0.76 ± 0.01 | **0.79 ± 0.01** | **0.80 ± 0.00** |
| LDA | 0.54 ± 0.00 | 0.55 ± 0.00 | 0.55 ± 0.00 | 0.56 ± 0.00 |
| SBERT pretrained | 0.69 ± 0.00 | 0.72 ± 0.00 | 0.75 ±0.01 | 0.75 ± 0.01 |
| Word2Vec pretrained | 0.72 ± 0.01 | 0.73 ± 0.01 | 0.74 ± 0.00 | 0.74 ± 0.01 |
| Word2Vec (d = 300) | 0.76 ± 0.00 | 0.76 ± 0.00 | 0.77 ± 0.00 | 0.77 ± 0.01 |
| Word2Vec (d = 64) | 0.76 ± 0.01 | 0.76 ± 0.00 | 0.76 ± 0.00 | 0.77 ± 0.00 |
| Doc2Vec pretrained | 0.73 ± 0.00 | 0.75 ± 0.00 | 0.76 ± 0.00 | 0.76 ± 0.00 |
| Doc2Vec (d = 300) | 0.55 ± 0.01 | 0.56 ± 0.00 | 0.57 ± 0.00 | 0.58 ± 0.00 |
| Doc2Vec (d = 64) | 0.54 ± 0.01 | 0.54 ± 0.00 | 0.55 ± 0.00 | 0.55 ± 0.00 |
| Sent2Vec pretrained | 0.73 ± 0.00 | 0.75 ± 0.00 | 0.77 ± 0.01 | 0.77 ± 0.01 |
| Sent2Vec (d = 600) | **0.77 ± 0.00** | **0.78 ± 0.00** | **0.79 ± 0.00** | 0.79 ± 0.01 |
| Sent2Vec (d = 64) | **0.77 ± 0.01** | **0.78 ± 0.00** | 0.78 ± 0.00 | 0.78 ± 0.00 |
| Ernie pretrained | 0.70 ± 0.01 | 0.71 ± 0.00 | 0.71 ± 0.00 | 0.73 ± 0.00 |

**Note:**
The best values with respect to confidence intervals are highlighted in bold.

labeled nodes is small (5% or 10%). When the share of labeled data is more significant, they show performance similar to Bag of Words. Opposite to the Cora results, one can note that pre-trained versions of all models substantially outperform their counterparts trained from scratch. The explanation for that might be that text length is relatively short, and the amount of data is not enough to restore dependencies between words.

Table 3 presents the results of text embedding methods on DBLP dataset. The results appear to be quite similar to the ones achieved on CiteSeer-M10 dataset: Bag of Words and TF-IDF perform equally well, with the former performing slightly better. Also, there is no dramatic degradation in the score when the percentage of labeled nodes is small.

Regarding advanced methods on DBLP dataset one can see that Sent2Vec outperforms all other architectures. Word2Vec also show very decent results (especially in terms of stability over different train rate sample). For Doc2Vec (in opposite to Sent2Vec) pre-trained version performed far better than the one trained from scratch. Again, advanced embeddings outperform classic techniques for a small percentage of labeled data and perform almost as good in case of more labeled data.

To sum up text embeddings experiments:

1. Advanced text embedding techniques such as Sent2Vec, Doc2Vec, Word2Vec outperform classic approaches such as Bag of Words and TF-IDF when the percentage of labeled data is small (<30%) and performs similarly when it is higher. LDA generally shows bad performance on all three datasets. Although pre-trained SBERT model shows decent results on CiteSeer-M10 and DBLP, it was outperformed by other architectures and even classic approaches.

2. Ernie framework generally shows poor performance. The reasons for that is that it is not optimized to produce sensible sentence embeddings. Architecture similar to Sentence Bert might be applied to improve the model quality for these kinds of tasks.

3. In general, advanced embedding techniques and LDA show very consistent results even for a small percentage of trained labels. In contrast, Bag of Words and TF-IDF show degrading results when a small share of labeled nodes is available. However, when text information is short (only titles) and there is more trained data in terms of the number of nodes (documents), this effect is mitigated.

4. TF-IDF and Bag of Words generally performs better for short texts (paper titles) because they are basically set of keywords. Advanced methods show good performance in both settings (short and long texts).

5. One can see that in some cases, pre-trained models perform better and in some cases worse, so it is better to experiment with both of the approaches.

### Network methods

According to the results on Cora dataset (Table 4) DeepWalk and Node2Vec show similar performance with DeepWalk being slightly better when the percentage of labeled nodes is larger than 5%. HOPE shows inferior results (near to random) for node classification task. When comparing the results with text embedding techniques (Table 1), one can note that DeepWalk and Node2Vec outperform all the other algorithms at a significant margin. Moreover, the tendency holds for different values of labeled nodes. It generally means that the Cora graph structure has a higher correlation with the target.

For Citeseer-M10 dataset (Table 4), DeepWalk and Node2Vec show identical performance for all labeled nodes' values, whereas HOPE again performs quite poorly. Interestingly, in contrast with Cora, here, one can see that text embedding techniques outperform network embeddings.

Table 4 shows the results for network embedding methods on DBLP dataset. Similar to the Citeseer-M10 dataset, we can see that DeepWalk and Node2Vec perform equally. Also, one can see that text embedding techniques severely outperform network embeddings on this dataset.

Generally, different datasets show different levels of importance for text and network data. For some datasets, nodes from the same class tend to link each other (the phenomenon is called homophily *Barabási & Pósfai (2016)*), which means that graph structure is beneficial for predicting the target. For other datasets, nodes might also tend to cite nodes from other classes. In this case, network information is less useful. Even though on some datasets, one type of information (text or network) significantly outperforms the other, still both might be useful as they tend to provide complementary information (different "views" on target).

### Fusion methods

When analyzing the results on the Cora dataset (Table 5), one can note that a naive combination of textual and network features performs similarly to more advanced

**Table 4 Network methods for node classification (micro-F1, metric lies between (0,1) and higher value means better results).**

| % Labels | 5% | 10% | 30% | 50% |
|---|---|---|---|---|
| *CORA* | | | | |
| DeepWalk | 0.72 ± 0.01 | **0.77 ± 0.00** | **0.81 ± 0.00** | **0.82 ± 0.01** |
| Node2Vec | **0.74 ± 0.01** | 0.76 ± 0.01 | 0.80 ± 0.00 | 0.81 ± 0.01 |
| HOPE | 0.29 ± 0.00 | 0.30 ± 0.00 | 0.30 ± 0.00 | 0.31 ± 0.00 |
| *CITESEER* | | | | |
| DeepWalk | **0.63 ± 0.00** | **0.65 ± 0.01** | **0.67 ± 0.00** | **0.68 ± 0.00** |
| Node2Vec | **0.63 ± 0.01** | **0.65 ± 0.00** | **0.67 ± 0.00** | **0.68 ± 0.00** |
| HOPE | 0.12 ± 0.00 | 0.13 ± 0.00 | 0.17 ± 0.00 | 0.20 ± 0.00 |
| *DBLP* | | | | |
| DeepWalk | **0.52 ± 0.00** | **0.53 ± 0.00** | **0.53 ± 0.00** | **0.53 ± 0.00** |
| Node2Vec | **0.52 ± 0.00** | **0.53 ± 0.00** | **0.53 ± 0.00** | **0.53 ± 0.00** |
| HOPE | 0.29 ± 0.01 | 0.30 ± 0.01 | 0.31 ± 0.00 | 0.31 ± 0.00 |

Note:
The best values with respect to confidence intervals are highlighted in bold.

**Table 5 Fusion methods on Cora for node classification (micro-F1, metric lies between (0,1) and higher value means better results).**

| % Labels | 5% | 10% | 30% | 50% |
|---|---|---|---|---|
| BoW + DeepWalk | 0.74 ± 0.01 | 0.80 ± 0.01 | 0.84 ± 0.00 | 0.86 ± 0.01 |
| Sent2Vec + DeepWalk | 0.76 ± 0.01 | 0.79 ± 0.00 | 0.84 ± 0.01 | 0.85 ± 0.01 |
| TADW - TF-IDF | 0.72 ± 0.02 | 0.80 ± 0.01 | 0.85 ± 0.01 | 0.86 ± 0.01 |
| TADW - Sent2Vec | 0.75 ± 0.01 | 0.80 ± 0.01 | 0.83 ± 0.00 | 0.85 ± 0.00 |
| TADW - Ernie | 0.57 (±0.02) | 0.69 (±0.01) | 0.80 (±0.00) | 0.82 (±0.00) |
| TriDNR | 0.59 ± 0.01 | 0.68 ± 0.00 | 0.75 ± 0.01 | 0.78 ± 0.01 |
| GCN - TF-IDF | 0.80 ± 0.01 | 0.83 ± 0.01 | 0.86 ± 0.01 | 0.87 ± 0.01 |
| GCN - Sent2Vec | 0.77 ± 0.01 | 0.82 ± 0.00 | 0.85 ± 0.01 | 0.87 ± 0.01 |
| GCN - Ernie | 0.60 ± 0.01 | 0.67 ± 0.02 | 0.77 ± 0.01 | 0.81 ± 0.00 |
| GAT - TF-IDF | **0.82 ± 0.02** | **0.84 ± 0.01** | **0.87 ± 0.01** | **0.88 ± 0.00** |
| GAT - Sent2Vec | 0.78 ± 0.00 | 0.81 ± 0.00 | 0.85 ± 0.01 | 0.86 ± 0.00 |
| GAT - Ernie | 0.58 ± 0.02 | 0.62 ± 0.02 | 0.71 ± 0.00 | 0.73 ± 0.00 |
| GraphSAGE - TF-IDF | 0.80 ± 0.01 | **0.84 ± 0.00** | **0.87 ± 0.01** | 0.87 ± 0.01 |
| GraphSAGE - Sent2Vec | 0.75 ± 0.01 | 0.80 ± 0.01 | 0.86 ± 0.01 | **0.88 ± 0.00** |
| GraphSAGE - Ernie | 0.29 ± 0.04 | 0.33 ± 0.05 | 0.34 ± 0.04 | 0.37 ± 0.02 |
| GIC - TF-IDF | 0.74 ± 0.01 | 0.81 ± 0.00 | 0.85 ± 0.00 | **0.88 ± 0.00** |
| GIC - Sent2Vec | 0.66 ± 0.00 | 0.76 ± 0.02 | 0.84 ± 0.00 | 0.86 ± 0.00 |
| GIC - Ernie | 0.34 ± 0.03 | 0.37 ± 0.02 | 0.37 ± 0.01 | 0.38 ± 0.01 |

Note:
The best values with respect to confidence intervals are highlighted in bold.

approaches TADW and TriDNR. Also, all approaches except TriDNR perform better than methods that use only text or network information, so one can conclude that these two types of information are complementing each other. GNNs significantly outperforms all

**Table 6 Fusion methods on Citeseer-M10 for node classification (micro-F1, metric lies between (0,1) and higher value means better results).**

| % Labels | 5% | 10% | 30% | 50% |
|---|---|---|---|---|
| BoW + DeepWalk | 0.73 ± 0.01 | 0.76 ± 0.00 | 0.81 ± 0.01 | 0.83 ± 0.01 |
| Sent2Vec + DeepWalk | 0.73 ± 0.01 | 0.75 ± 0.00 | 0.79 ± 0.01 | 0.80 ± 0.01 |
| TADW - TF-IDF | 0.47 ± 0.02 | 0.51 ± 0.01 | 0.57 ± 0.01 | 0.59 ± 0.01 |
| TADW - Sent2Vec | 0.57 ± 0.01 | 0.60 ± 0.00 | 0.65 ± 0.01 | 0.66 ± 0.01 |
| TADW - Ernie | 0.41 (±0.01) | 0.46 (±0.01) | 0.53 (±0.01) | 0.56 (±0.01) |
| TriDNR | 0.63 ± 0.01 | 0.68 ± 0.00 | 0.74 ± 0.01 | 0.77 ± 0.01 |
| GCN - TF-IDF | 0.71 ± 0.01 | 0.76 ± 0.01 | 0.81 ± 0.01 | 0.83 ± 0.01 |
| GCN - Sent2Vec | 0.73 ± 0.01 | **0.80 ± 0.00** | 0.84 ± 0.01 | **0.87 ± 0.01** |
| GCN - Ernie | 0.71 ± 0.01 | 0.75 ± 0.00 | 0.78 ± 0.00 | 0.79 ± 0.00 |
| GAT - TF-IDF | 0.72 ± 0.01 | 0.76 ± 0.01 | 0.82 ± 0.00 | 0.84 ± 0.01 |
| GAT - Sent2Vec | **0.75 ± 0.01** | 0.79 ± 0.00 | 0.81 ± 0.00 | 0.83 ± 0.00 |
| GAT - Ernie | 0.70 ± 0.02 | 0.74 ± 0.00 | 0.77 ± 0.00 | 0.78 ± 0.01 |
| GraphSAGE - TF-IDF | 0.72±0.01 | 0.77 ± 0.01 | 0.83 ± 0.00 | 0.85 ± 0.01 |
| GraphSAGE - Sent2Vec | **0.75 ± 0.01** | **0.80 ± 0.01** | **0.85 ± 0.00** | 0.86 ± 0.00 |
| GraphSAGE - Ernie | 0.58 ± 0.1 | 0.63 ± 0.01 | 0.65 ± 0.01 | 0.68 ± 0.01 |
| GIC - TF-IDF | 0.66 ± 0.00 | 0.70 ± 0.01 | 0.80 ± 0.00 | 0.83 ± 0.01 |
| GIC - Sent2Vec | 0.74 ± 0.01 | 0.78 ± 0.00 | 0.83 ± 0.00 | 0.84 ± 0.00 |
| GIC - Ernie | 0.49 ± 0.05 | 0.57 ± 0.02 | 0.57 ± 0.02 | 0.63 ± 0.00 |

**Note:**
The best values with respect to confidence intervals are highlighted in bold.

the other approaches. The best variation is the GAT with TF-IDF text encoding. Nevertheless, almost all other GNN approaches lead to similar solid results. We can observe an intriguing effect that the GIC model highly relies on the train set size. This model accounts for network substructures like communities. So to efficiently learn it, GIC requires substantial part of the graph to train to good quality. Another remarkable thing is that text embeddings with high individual performance require a larger subsample of a graph to achieve competitive results. However, this effect is less noticeable for the GAT model because GAT relies more on the attention mechanism than the graph structure.

Tables 6 and 7 show that on Citeseer and DBLP networks, unlike the Cora dataset, the naive combination of BoW and DeepWalk significantly outperforms much more advanced algorithms. However, some GNN models still show superior performance. GraphSAGE with Sent2Vec initial features gives the best results on almost all percentages of training nodes except for 50%. It refers us to the nature of GraphSAGE. It works in an inductive and scalable manner by sampling the node neighbors before GCN aggregation, so for larger networks than Cora, it performs better. However, GCN still shows good performance for large train parts. The difference is that GCN requires a whole node neighborhood while GraphSAGE samples it with random walks. It seems that this effect is similar to the dramatic growth of the GIC performance described above. Generally, the fusion of text and graph information shows superior results to network or text embeddings alone. However, GNN tends to perform better with sparse input features like TF-IDF.

**Table 7 Fusion methods on DBLP for node classification (micro-F1, metric lies between (0,1) and higher value means better results).**

| % Labels | 5% | 10% | 30% | 50% |
|---|---|---|---|---|
| BoW + DeepWalk | 0.77 ± 0.02 | 0.79 ± 0.01 | 0.81 ± 0.01 | 0.82 ± 0.01 |
| Sent2Vec + DeepWalk | 0.78 ± 0.01 | **0.80 ± 0.00** | 0.80 ± 0.01 | 0.80 ± 0.01 |
| TriDNR | 0.72 ± 0.01 | 0.75 ± 0.00 | 0.78 ± 0.01 | 0.79 ± 0.01 |
| GCN - TF-IDF | 0.71 ± 0.01 | 0.76 ± 0.01 | 0.81 ± 0.01 | **0.83 ± 0.01** |
| GCN - Sent2Vec | 0.78 ± 0.01 | **0.80 ± 0.00** | 0.81 ± 0.01 | 0.81 ± 0.01 |
| GCN - Ernie | 0.74 ± 0.01 | 0.75 ± 0.01 | 0.76 ± 0.01 | 0.77 ± 0.01 |
| GAT - TF-IDF | **0.79 ± 0.00** | **0.80 ± 0.00** | **0.82 ±0.00** | 0.82 ± 0.00 |
| GAT - Sent2Vec | **0.79 ± 0.00** | 0.79 ± 0.00 | 0.80 ± 0.01 | 0.80 ± 0.00 |
| GAT - Ernie | 0.73 ± 0.00 | 0.73 ± 0.00 | 0.75 ± 0.00 | 0.75 ± 0.00 |
| GraphSAGE - TF-IDF | **0.79 ± 0.01** | 0.79 ± 0.01 | 0.81 ± 0.00 | 0.82 ± 0.00 |
| GraphSAGE - Sent2Vec | **0.79 ± 0.00** | **0.80 ± 0.00** | 0.81 ± 0.00 | 0.81 ± 0.00 |
| GraphSAGE - Ernie | 0.70 ± 0.03 | 0.70 ± 0.02 | 0.71 ± 0.01 | 0.72 ± 0.01 |
| GIC - TF-IDF | 0.75 ± 0.00 | 0.77 ± 0.00 | 0.80 ± 0.00 | 0.81 ± 0.00 |
| GIC - Sent2Vec | 0.78 ± 0.00 | 0.79 ± 0.00 | 0.81± 0.00 | 0.81 ± 0.00 |
| GIC - Ernie | 0.51 ± 0.04 | 0.57 ± 0.02 | 0.63 ± 0.03 | 0.71 ± 0.01 |

**Note:**
The best values with respect to confidence intervals are highlighted in bold.

It could be because high-quality, dense vectors are susceptible to any change, so it is hard to mix information from different domains based on dense vectors.

## Link prediction

### Text methods

For the task of link prediction, one can expect text embeddings to perform more consistently concerning train ratio in comparison with network embeddings. Because network embedding techniques "suffer" twice when the percentage of train data is decreasing: firstly, it affects the initial graph, so it is harder to learn embeddings themselves. Secondly, it is harder to train a classifier using less data, whereas text embeddings have only the second problem: they are not dependent on the graph structure.

Table 8 shows results of text embedding techniques on Cora dataset. Again, one can see that BoW outperforms other methods, but LDA demonstrates much better performance for link prediction contrary to node classification. Similar to node classification problem on link prediction, advanced network embeddings perform worse when the percentage of train data is high but show better results when it gets lower.

Table 9 show results for classic methods for link prediction problem on CiteSeer dataset. Surprisingly, here BoW and TF-IDF perform very poorly, whereas SBERT shows superior performance. SBERT performance makes more sense since it is trained to differentiate two texts from each other, so it fits well for such tasks.

Generally, one can say that for link prediction choice of the text embeddings algorithm should be dependent on dataset, as there is no universal best performer.

**Table 8 Text embeddings on Cora for link prediction (micro-F1, metric lies between (0,1) and higher value means better results).**

| % Train edges | 5% | 10% | 30% | 50% |
|---|---|---|---|---|
| BoW | 0.69 ± 0.01 | 0.71 ± 0.00 | **0.75 ± 0.01** | **0.76 ± 0.00** |
| TF-IDF | 0.67 ± 0.01 | 0.69 ± 0.01 | 0.72 ± 0.01 | 0.74 ± 0.01 |
| LDA | 0.68 ± 0.01 | 0.69 ± 0.01 | 0.71 ± 0.01 | 071 ± 0.01 |
| SBERT pretrained | 0.69 ± 0.00 | 0.71 ± 0.00 | 0.74 ± 0.01 | **0.76 ± 0.01** |
| Word2Vec pretrained | 0.60 ± 0.02 | 0.62 ± 0.00 | 0.63 ± 0.00 | 0.64 ± 0.01 |
| Word2Vec (d = 300) | 0.68 ± 0.00 | 0.70 ± 0.00 | 0.72 ± 0.00 | 0.73 ± 0.01 |
| Word2Vec (d = 64) | 0.70 ± 0.00 | 0.70 ± 0.00 | 0.72 ± 0.01 | 0.73 ± 0.01 |
| Doc2Vec pretrained | 0.63 ± 0.02 | 0.66 ± 0.00 | 0.70 ± 0.00 | 0.70 ± 0.00 |
| Doc2Vec (d = 300) | 0.67 ± 0.01 | 0.70 ± 0.00 | 0.73± 0.00 | 0.74 ± 0.00 |
| Doc2Vec (d = 64) | 0.66 ± 0.01 | 0.68 ± 0.00 | 0.69 ± 0.00 | 0.69 ± 0.00 |
| Sent2Vec pretrained | 0.66 ± 0.01 | 0.69 ± 0.00 | 0.73 ± 0.00 | 0.75 ± 0.00 |
| Sent2Vec (d = 600) | **0.71 ± 0.00** | **0.72 ± 0.01** | **0.75 ± 0.00** | **0.76 ± 0.01** |
| Sent2Vec (d = 64) | 0.70 ± 0.01 | 0.71 ± 0.00 | 0.73 ± 0.00 | 0.74 ± 0.00 |
| Ernie pretrained | 0.56 ± 0.01 | 0.58 ± 0.01 | 0.62 ± 0.01 | 0.63 ± 0.01 |

Note:
The best values with respect to confidence intervals are highlighted in bold.

**Table 9 Text embeddings on Citeseer-M10 for link prediction (micro-F1, metric lies between (0,1) and higher value means better results).**

| % Train edges | 5% | 10% | 30% | 50% |
|---|---|---|---|---|
| BoW | 0.52 ± 0.01 | 0.52 ± 0.00 | 0.52 ± 0.01 | 0.52 ± 0.00 |
| TF-IDF | 0.52 ± 0.01 | 0.52 ± 0.01 | 0.53 ± 0.01 | 0.53 ± 0.00 |
| LDA | 0.69 ± 0.01 | 0.69 ± 0.01 | 0.70 ± 0.01 | 071 ± 0.01 |
| SBERT pretrained | **0.84 ± 0.00** | **0.85 ± 0.00** | **0.86 ± 0.01** | **0.86 ± 0.01** |
| Word2Vec pretrained | 0.53 ± 0.01 | 0.53 ± 0.01 | 0.54 ± 0.00 | 0.54 ± 0.01 |
| Word2Vec (d = 300) | 0.54 ± 0.00 | 0.54 ± 0.00 | 0.54 ± 0.00 | 0.54 ± 0.01 |
| Word2Vec (d = 64) | 0.54 ± 0.01 | 0.54 ± 0.01 | 0.54 ± 0.00 | 0.54 ± 0.01 |
| Doc2Vec (pretrained) | 0.55 ± 0.01 | 0.55 ± 0.00 | 0.55 ± 0.00 | 0.55 ± 0.00 |
| Doc2Vec (d = 300) | 0.77 ± 0.01 | 0.77 ± 0.00 | 0.78 ± 0.00 | 0.79 ± 0.00 |
| Doc2Vec (d = 64) | 0.77 ± 0.01 | 0.77 ± 0.01 | 0.77 ± 0.00 | 0.78 ± 0.00 |
| Sent2Vec pretrained | 0.54 ± 0.01 | 0.54 ± 0.01 | 0.55 ± 0.01 | 0.55 ± 0.01 |
| Sent2Vec (d = 600) | 0.54 ± 0.00 | 0.55 ± 0.01 | 0.55 ± 0.00 | 0.56 ± 0.01 |
| Sent2Vec (d = 64) | 0.53 ± 0.00 | 0.53 ± 0.01 | 0.54 ± 0.00 | 0.54 ± 0.01 |
| Ernie pretrained | **0.84 ± 0.01** | 0.84 ± 0.01 | 0.85 ± 0.01 | 0.85 ± 0.01 |

Note:
The best values with respect to confidence intervals are highlighted in bold.

## Network methods

When one masks edges of a network, it changes the graph structure (contrary to node classification), so it might be more challenging for structural network embedding method to perform well. Table 10 shows how network embedding algorithms perform on Cora for link prediction task.

**Table 10 Network embeddings for link prediction (micro-F1, metric lies between (0,1) and higher value means better results).**

| % Train Edges | 5% | 10% | 30% | 50% |
|---|---|---|---|---|
| *CORA* | | | | |
| DeepWalk | 0.56 ± 0.01 | 0.60 ± 0.00 | **0.66 ± 0.00** | 0.66 ± 0.01 |
| Node2Vec | **0.57 ± 0.01** | **0.61 ± 0.01** | 0.65 ± 0.01 | **0.68 ± 0.01** |
| HOPE | 0.50 ± 0.00 | 0.50 ± 0.00 | 0.51 ± 0.00 | 0.52 ± 0.00 |
| *CITESEER* | | | | |
| DeepWalk | **0.55 ±0.01** | **0.59 ± 0.01** | 0.66 ± 0.01 | 0.66 ± 0.00 |
| Node2Vec | **0.55 ± 0.00** | **0.59 ± 0.00** | **0.69 ± 0.00** | **0.71 ± 0.00** |
| HOPE | 0.50 ± 0.00 | 0.51 ± 0.00 | 0.54 ± 0.00 | 0.57 ± 0.00 |

Note:
The best values with respect to confidence intervals are highlighted in bold.

For Citeseer-M10 dataset (Table 10), the situation is quite similar to Cora dataset in a sense that Node2Vec performs better than DeepWalk, and both of these methods significantly outperform HOPE. Results for DBLP are omitted, but they are pretty much the same.

To sum these experiments up, for a link prediction problem (contrary to node classification), text information plays a much more critical role than graph structure.

### Fusion methods

For the link prediction task, one can see (Table 11) that classic methods (BoW and TF-IDF) outperform more advanced combinations (Sent2Vec and Word2Vec). Also, TADW performs on the same level that plain text embedding techniques. It might happen because TADW relies mostly on network information rather than on text. However, the performance of the GIC and SBERT mixture for this dataset is high. It means that methods preserving node clustering in the embedding space can improve the quality of each other.

A combination of TADW and Ernie shows by far the best results on the Citeseer-M10 dataset (Table 12). It becomes more evident when the percentage of training edges is high. It also follows previous results on using text embeddings, such as SBERT, on this dataset. Interestingly, usage of SBERT alone performs better than in combination with GCN. However, GraphSAGE mixture with SBERT shows close results to the pure text embedding. It means that the entire graph structure adds more noise to the fine embeddings for document clustering. Thus, careful selection of local neighbors is a crucial part to utilize all SBERT properties. Also, GNNs shows consistent results for different percentage of train edges, whereas other methods seem to degrade heavily as the percentage of train edges becomes lower.

Generally, one can see that for the link prediction task (as opposed to node classification problem), SBERT and Ernie show superior performance to other text embedding techniques when used alone and in combination with Graph Neural Networks. However, TF-IDF still shows high performance in fusion tasks.

**Table 11 Fusion embeddings on Cora for link prediction (micro-F1, metric lies between (0,1) and higher value means better results).**

| % Train Edges | 5% | 10% | 30% | 50% |
|---|---|---|---|---|
| TADW - BoW | 0.72 ± 0.02 | 0.72 ± 0.01 | 0.73 ± 0.01 | 0.73 ± 0.01 |
| TADW - TF-IDF | 0.73 ± 0.02 | 0.74 ± 0.01 | 0.74 ± 0.01 | 0.75 ± 0.01 |
| TADW - Sent2Vec | 0.70 ± 0.01 | 0.70 ± 0.01 | 0.71 ± 0.00 | 0.73 ± 0.00 |
| TADW - Word2Vec | 0.64 ± 0.01 | 0.68 ± 0.00 | 0.71 ± 0.01 | 0.72 ± 0.01 |
| TADW - Ernie | 0.51 ± 0.01 | 0.53 ± 0.01 | 0.54 ± 0.01 | 0.54 ± 0.01 |
| GCN - TF-IDF | **0.78 ± 0.01** | **0.78 ± 0.01** | **0.79 ± 0.01** | **0.80 ± 0.01** |
| GCN - Sent2Vec | 0.69 ± 0.01 | 0.71 ± 0.01 | 0.73 ± 0.01 | 0.75 ± 0.01 |
| GCN - SBERT | 0.67 ± 0.01 | 0.69 ± 0.01 | 0.71 ± 0.01 | 0.73 ± 0.01 |
| GCN (Custom) | 0.72 ± 0.01 | 0.75 ± 0.01 | 0.75 ± 0.01 | 0.75 ± 0.01 |
| GCN - Ernie | 0.62 ± 0.01 | 0.63 ± 0.00 | 0.63 ± 0.00 | 0.68 ± 0.01 |
| GAT - TF-IDF | 0.71 ± 0.01 | 0.73 ± 0.01 | 0.75 ± 0.01 | 0.75 ± 0.01 |
| GAT - Sent2Vec | 0.61 ± 0.01 | 0.61 ± 0.01 | 0.65 ± 0.01 | 0.68 ± 0.01 |
| GAT - SBERT | 0.65 ± 0.01 | 0.69 ± 0.01 | 0.72 ± 0.01 | 0.74 ± 0.01 |
| GAT - Ernie | 0.56 ± 0.01 | 0.56 ± 0.02 | 0.59 ± 0.01 | 0.62 ± 0.01 |
| GraphSAGE - TF-IDF | 0.75 ± 0.01 | **0.78 ± 0.01** | **0.79 ± 0.01** | **0.80 ± 0.01** |
| GraphSAGE - Sent2Vec | 0.66 ± 0.01 | 0.70 ± 0.01 | 0.74 ± 0.01 | 0.75 ± 0.01 |
| GraphSAGE - SBERT | 0.58 ± 0.01 | 0.62 ± 0.01 | 0.69 ± 0.01 | 0.64 ± 0.01 |
| GraphSAGE - Ernie | 0.50 ± 0.01 | 0.50 ± 0.01 | 0.53 ± 0.01 | 0.56 ± 0.01 |
| GIC - TF-IDF | 0.73 ± 0.01 | 0.75 ± 0.01 | 0.77 ± 0.01 | 0.78 ± 0.01 |
| GIC - Sent2Vec | 0.74 ± 0.01 | 0.75 ± 0.01 | 0.77 ± 0.01 | 0.78 ± 0.01 |
| GIC - SBERT | 0.74 ± 0.01 | 0.76 ± 0.01 | 0.78 ± 0.01 | **0.80 ± 0.01** |
| GIC - Ernie | 0.65 ± 0.01 | 0.69 ± 0.01 | 0.69 ± 0.01 | 0.74 ± 0.01 |

**Note:**
The best values with respect to confidence intervals are highlighted in bold.

## Network visualization

The main goal of network visualization is to produce a "meaningful" 2D plot of nodes. Meaningful visualization would place nodes from one class close to each other and nodes from different classes far away from each other. Consequently, having solved the visualization problem, one automatically gets node clustering and vice versa.

To produce a 2D plot, one has to find a vector of two points describing the position of a node. This problem can be solved in two ways using network embeddings:

1. Explicitly learn embeddings of size 2 using any methods described in the previous chapters.
2. First, learn embeddings of length $d$, then use a dimensionality reduction method to obtain vectors of the size 2.

We follow the second approach since the first one generally produces worse results, as it is a much more challenging task to learn a realistic representation of size 2.

For embeddings compression, we will use t-distributed Stochastic Neighbor Embedding (t-SNE) presented by *Maaten & Hinton (2008)*. Firstly, t-SNE initialize the projection to

**Table 12 Fusion embeddings on Citeseer-M10 for link prediction (micro-F1, metric lies between (0,1) and higher value means better results).**

| % Train Edges | 5 % | 10 % | 30 % | 50 % |
|---|---|---|---|---|
| TADW - BoW | 0.50 ± 0.01 | 0.51 ± 0.02 | 0.51 ± 0.01 | 0.52 ± 0.01 |
| TADW - TF-IDF | 0.51 ± 0.01 | 0.51 ± 0.01 | 0.51 ± 0.01 | 0.52 ± 0.01 |
| TADW - Sent2Vec | 0.52 ± 0.01 | 0.53 ± 0.00 | 0.53 ± 0.00 | 0.54 ± 0.00 |
| TADW - Word2Vec | 0.52 ± 0.01 | 0.52 ± 0.01 | 0.53 ± 0.00 | 0.53 ± 0.00 |
| TADW - Ernie | **0.74 ± 0.01** | **0.75 ± 0.01** | **0.77 ± 0.02** | 0.78 ± 0.01 |
| GCN - TF-IDF | 0.68 ± 0.01 | 0.69 ± 0.01 | 0.70 ± 0.01 | 0.70 ± 0.01 |
| GCN - Sent2Vec | 0.59 ± 0.01 | 0.62 ± 0.01 | 0.67 ± 0.01 | 0.68 ± 0.01 |
| GCN - SBERT | 0.68 ± 0.01 | 0.70 ± 0.01 | 0.72 ± 0.01 | 0.77 ± 0.01 |
| GCN (Custom) | 0.61 ± 0.01 | 0.67 ± 0.00 | 0.68 ± 0.01 | 0.68 ± 0.01 |
| GCN - Ernie | 0.67 ± 0.01 | 0.67 ± 0.01 | 0.76 ± 0.00 | 0.78 ± 0.01 |
| GAT - TF-IDF | 0.60 ± 0.01 | 0.63 ± 0.01 | 0.65 ± 0.01 | 0.64 ± 0.01 |
| GAT - Sent2Vec | 0.59 ± 0.01 | 0.63 ± 0.01 | 0.64 ± 0.01 | 0.63 ± 0.01 |
| GAT - SBERT | 0.61 ± 0.01 | 0.65 ± 0.01 | 0.71 ± 0.01 | 0.73 ± 0.01 |
| GAT - Ernie | 0.61 ± 0.00 | 0.64 ± 0.01 | 0.69 ± 0.01 | 0.70 ± 0.01 |
| GraphSAGE - TF-IDF | 0.66 ± 0.01 | 0.67 ± 0.01 | 0.73 ± 0.01 | 0.78 ± 0.01 |
| GraphSAGE - Sent2Vec | 0.64 ± 0.01 | 0.66 ± 0.01 | 0.73 ± 0.01 | 0.78 ± 0.01 |
| GraphSAGE - SBERT | 0.61 ± 0.01 | 0.63 ± 0.01 | 0.71 ± 0.01 | **0.83 ± 0.01** |
| GraphSAGE - Ernie | 0.63 ± 0.02 | 0.72 ± 0.01 | 0.72 ± 0.01 | 0.80 ± 0.01 |
| GIC - TF-IDF | 0.62 ± 0.01 | 0.66 ± 0.01 | 0.74 ± 0.01 | 0.80 ± 0.01 |
| GIC - Sent2Vec | 0.62 ± 0.01 | 0.66 ± 0.01 | 0.75 ± 0.01 | 0.81 ± 0.01 |
| GIC - SBERT | 0.63 ± 0.01 | 0.66 ± 0.01 | 0.75 ± 0.01 | 0.78 ± 0.01 |
| GIC - Ernie | 0.63 ± 0.01 | 0.66 ± 0.00 | 0.73 ± 0.01 | 0.81 ± 0.00 |

**Note:**
The best values with respect to confidence intervals are highlighted in bold.

the two-dimensional space. Then, it calculates similarities between all points in both spaces and converts it to the joint probability distributions. Finally, it enhances projection via minimization of the Kullback-Leibler divergence between distributions in original and manifold spaces.

For all models, hyperparameters are chosen in the same way as in previous experiments. For GCN model, we take the output of the first activation layer.

When analyzing result obtained without fusion (Fig. 1), one can note that even simple TF-IDF method provides a solid baseline as most of the classes (represented by colors) are clearly separable. There are some exceptions, such as purple and pink classes, which were not concentrated. Sent2Vec shows even better results: samples of every class are tightly clustered together. The only problem is that clusters themselves are very close to each other, which means that it would be rather hard to separate them using clustering algorithms. For DeepWalk model, different classes are placed quite far away from each other, which solves the problem mentioned above. However, some classes are split, for example, green and red classes consist of multiple clusters.

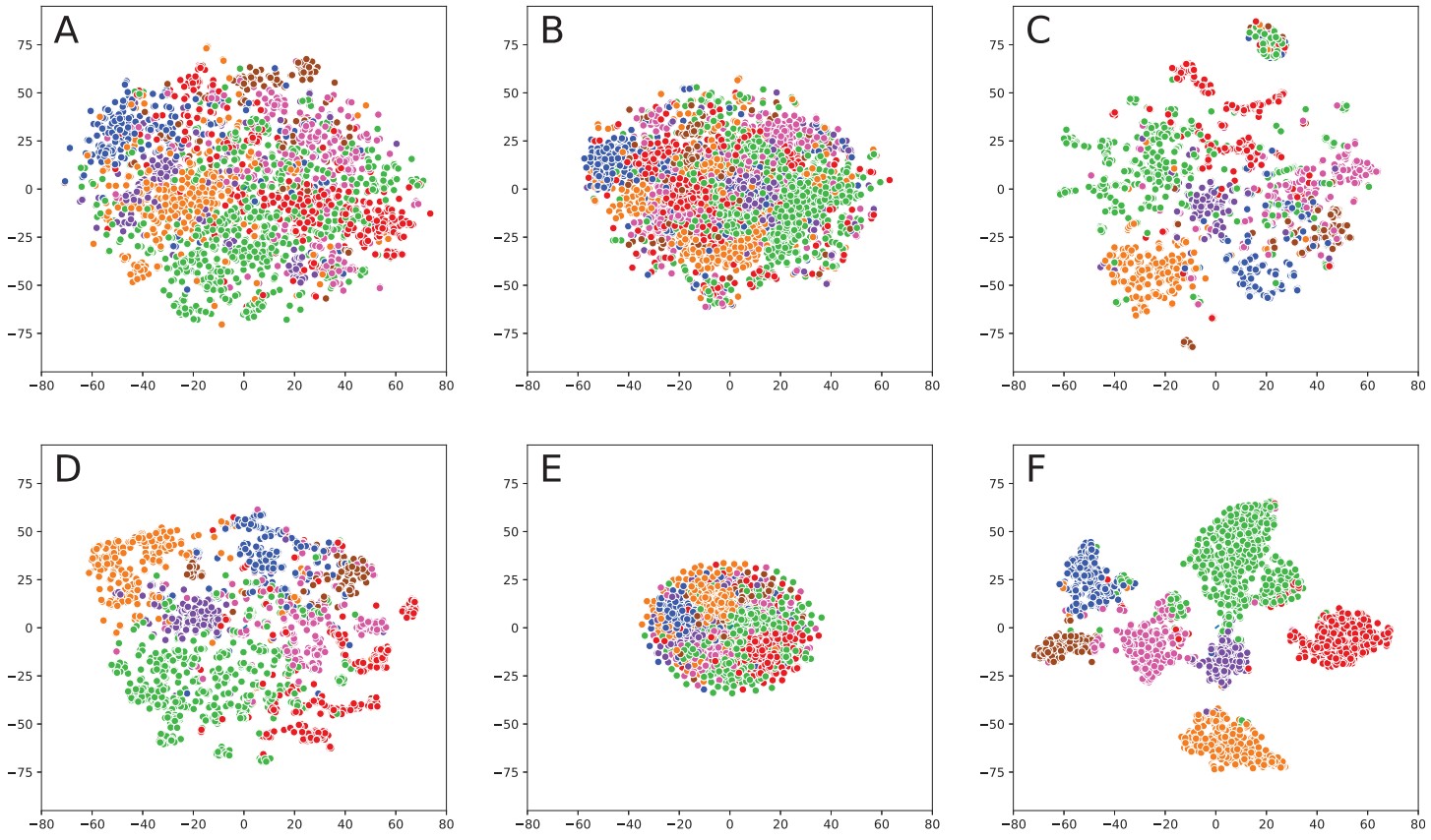

**Figure 1 Embeddings visualization on Cora.** (A) TF-IDF (text embedding). (B) Sent2Vec (text embedding). (C) DeepWalk (network embedding). (D) TADW (fusion). (E) TriDnr (fusion). (F) GCN (fusion).

Fusion methods provide even better visualizations: TADW shows the result that is very similar to DeepWalk but with better consistency across different classes. TriDNR is more similar to Sent2Vec because it provides a very clean separation of classes. However, the classes themselves are located very close to each other, so it would be hard to apply clustering algorithms. GCN provides the best result since classes are far away from each other and clearly separated, so it would be elementary to cluster points in the embedding space.

## DISCUSSION

One can see that fusion of text and graph information shows superior results for all machine learning tasks on graphs compared to methods that use only text or network information. It proves that text information and graph structure are complementary to each other, but each component's contribution depends on a task and a dataset. It is also clear that using advanced text embedding techniques such as Sent2Vec and SBERT can significantly boost the performance of fusion methods. The reason is that advanced text embeddings can better capture the semantic of the words (synonyms, antonyms) and, therefore, better generalize. Moreover, pre-trained embeddings might be preferable when the number of training samples is low. However, Bag of Words and TF-IDF provide a

substantial baseline for machine learning tasks on citation networks because a set of keywords can efficiently represent nodes (scientific papers). Also, sparse vectors are less sensible to the minor changes, so it fits the fusion task better.

The choice of the text embedding technique should be task-dependent: SBERT works better for link prediction, whereas Sent2Vec shows good performance for node classification. It can be explained by the fact that Sent2Vec is aimed to preserve the overall semantics of a text. In contrast, SBERT is specifically trained to predict whether two texts describe the same thing or not, so it is no wonder that SBERT is incredibly good at solving a link prediction problem. The GraphSAGE shows better performance for large networks compared to the other GNNs because it is designed for scalable inductive learning. GIC works much better when the training part of the network is large. It requires many details about graph substructure to utilize all power of cluster-level loss. Also, SBERT and ERNIE perform better in fusion when the GNN model accurately selects the nodes to be aggregated. So the modification of models like GIC and GraphSAGE, which work with subgraph structures, could leverage the boost in its performance after some modifications.

Unfortunately, our custom GCN architecture, which allows learning word and network embeddings simultaneously, does not outperform state-of-the-art algorithms. Nevertheless, there is some potential for it. As mentioned before, the GCN part could be replaced by GraphSAGE or GIC models. Furthermore, pretraining of text embedding layer with further fine-tuning can show much better results on bigger networks (millions of nodes) due to the better quality of source text embeddings. The other possible modification of the fusion technique is to use the graph as a source of pairs for the SBERT framework.

## CONCLUSION

A comprehensive comparison of different fusion methods for machine learning problems on networks was conducted in this work. The best combinations of network and text embeddings for different machine learning tasks were outlined and compared with the traditional approaches. The new GCN architecture was proposed for learning text and network representations simultaneously. Main conclusions of the work:

1. Fusion of text and graph information allows boosting performance on machine learning tasks significantly.
2. Usage of advanced text embeddings such as Sent2Vec and SBERT can improve the accuracy of different fusion architectures such as TADW and GCN. SBERT generally works better for link prediction, Sent2Vec for node classification.
3. There is no universal solution that fits all problems and all datasets. Different methods (and combinations of methods) might work better for different datasets.
4. Proposed GCN modification does not work well for datasets considered in this work but might show a better performance for bigger networks with more text data.

This work can be continued in the following ways. Firstly, it is promising to experiment with the proposed GCN architecture using bigger networks (ideally millions of nodes). It might show better results because a lot of data is required to learn sensible word embeddings. However, it is better to use models that work on the subgraph level (GraphSAGE or GIC) for better scalability and synergy with SBERT and ERNIE features. Another possible extension is to use a joint loss to learn network and text embeddings simultaneously. For instance, combining GNNs and BERT might present very competitive results for link prediction. Also, networks could be a source of positive pairs for contrastive learning technique (like SBERT is trained) or even provide more insights on knowledge graph related problems (like suggested by *Deng, Rangwala & Ning (2020)*).

## ACKNOWLEDGEMENTS

The authors are grateful to Daniil Tikhomirov for his invaluable contribution to polishing the article presentation and text.

### Funding

The article was prepared within the framework of the HSE University Basic Research Program. There was no additional external funding received for this study. The funders had no role in study design, data collection and analysis, decision to publish, or preparation of the manuscript.

### Grant Disclosures

The following grant information was disclosed by the authors:
HSE University Basic Research Program.

### Competing Interests

The authors declare that they have no competing interests.

### Author Contributions

- Ilya Makarov conceived and designed the experiments, analyzed the data, authored or reviewed drafts of the paper, and approved the final draft.
- Mikhail Makarov conceived and designed the experiments, performed the experiments, analyzed the data, performed the computation work, prepared figures and/or tables, authored or reviewed drafts of the paper, and approved the final draft.
- Dmitrii Kiselev conceived and designed the experiments, performed the experiments, analyzed the data, performed the computation work, prepared figures and/or tables, authored or reviewed drafts of the paper, and approved the final draft.

### Data Availability

Data and code are available at GitHub: https://github.com/MakarovIA/graph_text

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
