# Peer review of "Fusion of text and graph information for machine learning problems on networks"

_PeerJ Computer Science, doi:10.7717/peerj-cs.526_

## Round 0.1 · original submission · Major Revisions

Based on the reviewers’ comments and my own evaluation, I think the manuscript needs to be significantly improved before further consideration for publication. In particular, when revising your manuscript, please especially pay attention to the following issues.

Please carefully improve the writing style / English language. Both of the two reviewers have kindly pointed out very detailed suggestions.

Please re-organize the structure of the manuscript, in particular, according to the first reviewer’s comments.

Please enhance the introduction to the field background.

Reviewer 1 ·

Basic reporting

1. The writing should be improved. I suggest the authors edit the incomplete sentence at lines 313-314 "One can that the results are quite similar to ...". Meanwhile, there are many complex sentences where adding punctuations can make it clear. For example, the sentence at lines 254-256 "After that in order to obtain embeddings for sentences ...", and the sentence at lines 268-269 "After embeddings for all of the nodes are learned ...". There are still some minor issues, e.g., the first word of a sentence should be capitalized, as shown at lines 259 and 264.

2. The paper is not well-organized and the presentation should be improved. The authors use a very long section with lots of tables to explain the results of node classification and link classification in different settings. Readers may easily be overwhelmed by those tables and text. It would be better to provide some figures to make the comparison result clearer and digestible.

3. Insufficient field background. The literature introduction in Section 2 seems to be weak. For instance, the authors only introduce the algorithms being evaluated in their study and ignore some of the more recent and popular graph neural network models, such as GraphSage [1], GAT [2]. The authors are expected to provide detailed background knowledge about the network and text embedding methods and provide reasons for the model chosen in the experiment. Meanwhile, some studies are focusing on fusing network and text information, e.g., ERNIE[3], Glean[4]. It would be beneficial if the author could include some advanced fusion models when introducing the background.

[1] Hamilton, Will, Zhitao Ying, and Jure Leskovec. "Inductive representation learning on large graphs." Advances in neural information processing systems. 2017.
[2] Veličković, Petar, et al. "Graph attention networks." arXiv preprint arXiv:1710.10903 (2017).
[3] Zhang, Zhengyan, et al. "ERNIE: Enhanced language representation with informative entities." arXiv preprint arXiv:1905.07129 (2019).
[4] Deng, Songgaojun, Huzefa Rangwala, and Yue Ning. "Dynamic Knowledge Graph based Multi-Event Forecasting." Proceedings of the 26th ACM SIGKDD International Conference on Knowledge Discovery & Data Mining. 2020.

Experimental design

no comment

Validity of the findings

no comment

Additional comments

This work is an interesting empirical study that compares different fusion methods of network and text embeddings for machine learning tasks on graphs. The experimental settings are clearly provided, and the results seem to be detailed.

Reviewer 2 ·

Basic reporting

The English language should be improved to ensure that your international audience can clearly understand your paper. I would suggest avoiding informal language in the manuscript.
TYPOS:
Add parentheses and commas to all references in the paper. For example, in line 33: matrices Perozzi et al. (2014) Grover and Leskovec (2016) -> matrices (Perozzi et al. (2014), Grover and Leskovec (2016))
I would add some references to each mentioned problem in the first paragraph of the introduction;
Line 29: Its main idea to map -> Its main idea is to map;
Line 29: low dimensional space -> low-dimensional space;
Line 41: semantic of words, using their context -> semantic of words using their context;
Line 43: which makes it more powerful -> which makes this method more powerful;
Line 71: structure -> structures;
Line 79: from word -> from a word;
Line 84: an approach allow -> an approach allows;
Line 89: Doc2Vec additionally create -> Doc2Vec additionally creates;
Line 89: When target word -> When a target word;
Line 92: BERT Devlin et al. (2018) the main idea is -> The main idea of BERT (Devlin et al. (2018)) is. It is better to rephrase this sentence in full.
Line 94: instead if employs -> instead it employs;
Line 95: SBERT Reimers and Gurevych (2019) framework propose -> SBERT framework (Reimers and Gurevych (2019)) proposes;
Line 110: embeddings (and vice versa) -> embedding and vice versa;
Line 114: TADW Yang et al. (2015) algorithm -> TADW algorithm (Yang et al. (2015));
Line 145: It’s important -> it is important. “It’s” is informal English;
Line 196: and therefore show much -> and therefore, show much;
Line 202: a quite high model -> a quite high, model;
Line 203: Therefore it is -> Therefore, it is;
Line 204: However this is -> However, this is;
Line 205: and therefore can afford -> and therefore, can afford;
Line 246: in most -> In most;
Line 251: Also it interesting -> Also it is interesting;
Line 253: In this case padded -> In this case, padded;
Line 255: mean and max functions applied -> mean and max functions must be applied;
Line 259: firstly -> Firstly;
Line 273: the results is reported -> the results are reported;
Line 276: remove the upside down question mark;
Line 303: becomes small -> becomes smaller;
Line 313: One can that -> One can note that. It is better it replace it with some synonyme because you already used this phrase in the last paragraph;
Line 385: one mask edges -> one masks edges;
Line 421: Therefore the embeddings dimension size -> Therefore, the embedding dimension size;
Line 423: embeddings algorithms -> embedding algorithms;
Line 423: obtained embeddings are to be mapped -> obtained embeddings are mapped;
Line 430: results -> result;
Line 432: Deep Walk’s visualization -> Deep Walk method;
Line 435: TADW shows results that are very -> TADW show the result that is very;
Line 439: the best results -> the best result;

English must be significantly improved. My suggestions are following:
Line 18: a user people -> users;
Line 68: in order to be consistent use either “Word2Vec” as in the line 41 or “word2vec” everywhere in the paper;
Line 128: you have not specified anywhere in the paper for what the abbreviation LPP stands. Again, in order to be consistent either use node classification problem or Node Classification problem as in the line 22;
Line 129 and 130: the sentence “GCN simply applies the adjacency matrix...” is imprecise. You can not apply the adjacency matrix for something. So I would suggest to change this part and add more details to the explanation of GCN architecture;
Line 165: move the reference “Pan et al. (2016)” to the end of this sentence;
Line 195: it is unclear what LDA algorithm is. You did not mention it early in the paper, so it would be good to add some explanation.
Line 199: the sentence “When trained...” is not quite clear to me. Which two variations did you mean there? And what does “dimension equals to d and 64” mean?
Authors use a phrase “it is sensible to...” in the paper. I am not sure that it is the correct phrase. Instead, use: it makes sense/be worthwhile;
Line 205: replace “still” with more formal words such as however or nevertheless and start a new sentence.
Line 208: I would split this sentence and start a new sentence with “vector size...”;
Line 224: I believe instead of “three network embeddings” should be “three network embedding methods/techniques”;
Line 226: for me it is not clear in which settings the methods tend to outperform others;
Line 260: replace “After that” with a synonyme, since you use it too often in the paper;
Line 264: split into train test -> split into train set and test set / split into train and test sets;
Line 270: I guess regression classifier is trained on the learned embedding of graphs in the train set but not on the train set itself;
Line 291: the by far best model -> By far the best model;
Line 303: What is the absolute number of samples? I did not understand what the word “absolute” meant in this context;
Line 315: I would rephrase this sentence or at least not use the word “perform” twice in one sentence;
Line 317: DBLP dataset Table 6 one can see, DBLP dataset one can see in Table 6;
Line 318: for me it is unclear what you mean by stability. You could add some explanation;
Table 6: why did you bold the result in Sent2Vec pretrained/50%?
Line 337: as far as I understand here begins the results for graph embedding methods, so I would add some paragraph title. It is difficult to follow the paper;
Line 342: that for Cora network data -> that the Cora network. Moreover, the last sentence is hard to follow, so I would rephrase it in full.
Line 356: Aain I would add a paragraph title to separate results for graph embedding methods, text embedding methods and fusion methods.
Line 385: the same, add a paragraph title.
Line 393: the same, add a paragraph title, for example, “Result for Link Prediction problem.”
Line 398: the sentence “It is quite predictable...” is not clear for me.
Line 399: I would not use the word “though” in the scientific paper;
Line 422: I would use the word “entire” instead of “whole”, since it is more formal;
Line 424: add a reference for T-SNE algorithm;
Line 426: “In the case of GCN activations of the first convolutional layer” is not fully clear. What are GCN activations? Did you mean activation functions?
Line 429: Again, I would not use “though” in the paper. You can write: “However, there are some exceptions...”
Line 439 and 440: it would be very easy to make a clusterization -> it would be very easy to cluster points.
Line 448: “what is more” is a more informal way of saying, so use formal synonyms such as moreover, furthermore.
The last sentence in the discussion is too long. I would split it into two or three sentences.

Experimental design

In general, experiments are designed in a good way but sometimes experiments lack some details.

In the Validation section, I would add more details about what your input data is. Moreover, in line 260 you split your data into train and test sets for Node Classification problem and then in your tables you consider different percentages of node labels. So I think you could add in Section 3.5.1 that you consider different ratios of node labels and the same for Link Prediction problem;
In line 259 it is stated that the input data is preprocessed. How did you preprocess data? Is this preprocessing step described in Section 3.2 for text data? Or did you preprocess differently? If yes, then it would be better to add details. But then I did not find how you preprocessed the graph-structured data. Also, I did not understand the part “nodes of the network are mapped into latent space using the input information”. What kind of input information exactly? Do you mean the input data which you mentioned at the beginning of this sentence?
Line 269: What are xu and xvin your case? Add more details;

Validity of the findings

The experiments compare the accuracy of different fusion methods which allows the audience to comprehend how the text information could impact the results of node classification and link prediction problems.
The data, which is used in the paper, is well-known benchmark graph-structured data available online.
All conclusions are linked to original research questions and results are well-described.

---

## Round 0.2 · Minor Revisions

Please carefully improve the entire text and then re-submit.

Reviewer 1 ·

Basic reporting

I suggest that the authors do another proofreading to improve the text. Most importantly, articles (i.e., a, an, and the) are largely abused in this paper, please correct these grammar issues.

Here are some obvious typos in Sections 1 and 2:
Line 18: the(a) great variety of
Line 45: (A) More advanced approach
Line 73: a(remove) text information
Line 78: remove "."
Line 107: learn to embed -> learns embeddings
Line 110: (a) more efficient, (the) random walk idea
Line 115: (an) important property
Line 129: to use them(remove)
Line 137: the algorithm learn(s)
Line 142: link prediction problem(s) or node classification(s)
Line 147: Graph Attention n(N)etworks (GAT), (the) self-attention mechanism

Experimental design

The experimental design is good, with detailed settings.

Validity of the findings

The results seem to be comprehensive, and the authors present detailed explanations.

Reviewer 2 ·

Basic reporting

The paper was significantly improved. I like the structure of the paper now. It is easy to read the paper and follow the main ideas.

I have some minor remarks but in general, I think, the paper can be accepted:

- I would use a word “graph: everywhere throughout the paper instead of mixing “graph” and “network” words.
- Everywhere in the paper, you use "the loss" such as in line 56 or 155. Instead, I would use "the loss function".
Line 65: and models choice -> and the choice of models;
Line 73: focus is in -> focus is on;
Line 154: GIC leverage - > GIC leverages;
Line 178: the better the quality -> the better the quality is.
Line 277: I would use either logistic regression or Logistic Regression as in the line 394.

Experimental design

Now all methods are described in a good clear way.

Validity of the findings

All results of the experiments are analyzed clearly and with sufficient details.

---

## Author Rebuttal · Round 0.2

Dear PeerJ CS Editors,

We thank the Reviewers for their generous and fruitful comments on the manuscript. We have placed Reviewers' comments below and answered their concerns in a point-by-point manner, starting with "">> Review answering text" and also reflecting updated text line numbers in a new Manuscript.

In particular, all the code supporting the Experiments section that was attached to the original submission has been updated and attached to the Revised Manuscript in Supplementary files.

We believe that the Manuscript is now suitable to be published in PeerJ CS.

Sincerely yours,
On behalf of all authors.

==================================================================

# Editor comments

Please carefully improve the writing style / English language. Both of the two reviewers have kindly pointed out very detailed suggestions.

Please re-organize the structure of the manuscript, in particular, according to the first reviewer's comments.

Please enhance the introduction to the field background.

==================================================================

# Reviewer 1 (Anonymous)

Basic reporting
1. The writing should be improved. I suggest the authors edit the incomplete sentence at lines 313-314 "One can that the results are quite similar to ...". Meanwhile, there are many complex sentences where adding punctuations can make it clear. For example, the sentence at lines 254-256 "After that in order to obtain embeddings for sentences ...", and the sentence at lines 268-269 "After embeddings for all of the nodes are learned ...". There are still some minor issues, e.g., the first word of a sentence should be capitalized, as shown at lines 259 and 264.
>> *Thanks for the comments, the text is seriously proofread, mistakes are fixed.*

2. The paper is not well-organized and the presentation should be improved. The authors use a very long section with lots of tables to explain the results of node classification and link classification in different settings. Readers may easily be overwhelmed by those tables and text. It would be better to provide some figures to make the comparison result clearer and digestible.

>> *We have added structure details for each section and reorganize Experiment and Discussion sections in order to provide a consistent way of describing results across the paper. We believe, now, it should be much easier to follow the experiments. We also merged Tables corresponding to common tasks.*

3. Insufficient field background. The literature introduction in Section 2 seems to be weak. For instance, the authors only introduce the algorithms being evaluated in their study and ignore some of the more recent and popular graph neural network models, such as GraphSage [1], GAT [2]. The authors are expected to provide detailed background knowledge about the network and text embedding methods and provide reasons for the model chosen in the experiment. Meanwhile, some studies are focusing on fusing network and text information, e.g., ERNIE[3], Glean[4]. It would be beneficial if the author could include some advanced fusion models when introducing the background.

[1] Hamilton, Will, Zhitao Ying, and Jure Leskovec. "Inductive representation learning on large graphs." Advances in neural information processing systems. 2017.
[2] Veličković, Petar, et al. "Graph attention networks." arXiv preprint arXiv:1710.10903 (2017).
[3] Zhang, Zhengyan, et al. "ERNIE: Enhanced language representation with informative entities." arXiv preprint arXiv:1905.07129 (2019).
[4] Deng, Songgaojun, Huzefa Rangwala, and Yue Ning. "Dynamic Knowledge Graph based Multi-Event Forecasting." Proceedings of the 26th ACM SIGKDD International Conference on Knowledge Discovery & Data Mining. 2020.

>> *We have added experiments and its discussion for more advanced Graph Neural Networks. It follows a similar pipeline to the existing GCN. Moreover, as one of the textual encoding approaches, we add ERNIE and validate its mixture with other models. . We did not include experiments with Glean, because it is used mostly for Knowledge Graph related task which are far from our experiments, however, we cite Glean as a general fusion model.*

Experimental design

no comment

Validity of the findings
no comment

## Comments for the Author

This work is an interesting empirical study that compares different fusion methods of network and text embeddings for machine learning tasks on graphs. The experimental settings are clearly provided, and the results seem to be detailed.

===================================================================

# Reviewer 2 (Anonymous)

Basic reporting
The English language should be improved to ensure that your international audience can clearly understand your paper. I would suggest avoiding informal language in the manuscript.

>> *Here and below we thank you for the giant work on our text, thank you very much for pointing out these inconsistencies. We have fixed them and proofread the paper several times, we hope now it is much more clear from both language and style. Below we will answer some of the questions, corrected in the text.*

**TYPOS:**
Add parentheses and commas to all references in the paper. For example, in line 33: matrices Perozzi et al. (2014) Grover and Leskovec (2016) -> matrices (Perozzi et al. (2014), Grover and Leskovec (2016))
I would add some references to each mentioned problem in the first paragraph of the introduction;
Line 29: Its main idea to map -> Its main idea is to map;
Line 29: low dimensional space -> low-dimensional space;
Line 41: semantic of words, using their context -> semantic of words using their context;
Line 43: which makes it more powerful -> which makes this method more powerful;
Line 71: structure -> structures;
Line 79: from word -> from a word;
Line 84: an approach allow -> an approach allows;
Line 89: Doc2Vec additionally create -> Doc2Vec additionally creates;
Line 89: When target word -> When a target word;
Line 92: BERT Devlin et al. (2018) the main idea is -> The main idea of BERT (Devlin et al. (2018)) is. It is better to rephrase this sentence in full.

Line 94: instead if employs -> instead it employs;
Line 95: SBERT Reimers and Gurevych (2019) framework propose -> SBERT framework (Reimers and Gurevych (2019)) proposes;
Line 110: embeddings (and vice versa) -> embedding and vice versa;
Line 114: TADW Yang et al. (2015) algorithm -> TADW algorithm (Yang et al. (2015));
Line 145: It's important -> it is important. "It's" is informal English;
Line 196: and therefore show much -> and therefore, show much;
Line 202: a quite high model -> a quite high, model;
Line 203: Therefore it is -> Therefore, it is;
Line 204: However this is -> However, this is;
Line 205: and therefore can afford -> and therefore, can afford;
Line 246: in most -> In most;
Line 251: Also it interesting -> Also it is interesting;
Line 253: In this case padded -> In this case, padded;
Line 255: mean and max functions applied -> mean and max functions must be applied;
Line 259: firstly -> Firstly;
Line 273: the results is reported -> the results are reported;
Line 276: remove the upside down question mark;
Line 303: becomes small -> becomes smaller;
Line 313: One can that -> One can note that. It is better it replace it with some synonyme because you already used this phrase in the last paragraph;
Line 385: one mask edges -> one masks edges;
Line 421: Therefore the embeddings dimension size -> Therefore, the embedding dimension size;
Line 423: embeddings algorithms -> embedding algorithms;
Line 423: obtained embeddings are to be mapped -> obtained embeddings are mapped;
Line 430: results -> result;
Line 432: Deep Walk's visualization -> Deep Walk method;
Line 435: TADW shows results that are very -> TADW show the result that is very;
Line 439: the best results -> the best result;

**English must be significantly improved. My suggestions are following:**
Line 18: a user people -> users;
Line 68: in order to be consistent use either "Word2Vec" as in the line 41 or "word2vec" everywhere in the paper;
Line 128: you have not specified anywhere in the paper for what the abbreviation LPP stands. Again, in order to be consistent either use node classification problem or Node Classification problem as in the line 22;

*>> LPP stands for Link Prediction Problem, added definition. All the writings now should be consistent across the paper.*

Line 129 and 130: the sentence "GCN simply applies the adjacency matrix..." is imprecise. You can not apply the adjacency matrix for something. So I would suggest to change this part and add more details to the explanation of GCN architecture;

Line 165: move the reference "Pan et al. (2016)" to the end of this sentence;

Line 195: it is unclear what LDA algorithm is. You did not mention it early in the paper, so it would be good to add some explanation.

*>> Added paragraph to related work.*

Line 199: the sentence "When trained..." is not quite clear to me. Which two variations did you mean there? And what does "dimension equals to d and 64" mean?

*>> We either use the size of pretrained model or 64 as a size, explained in details in the text.*

Authors use a phrase "it is sensible to..." in the paper. I am not sure that it is the correct phrase. Instead, use: it makes sense/be worthwhile;

Line 205: replace "still" with more formal words such as however or nevertheless and start a new sentence.

Line 208: I would split this sentence and start a new sentence with "vector size...";

Line 224: I believe instead of "three network embeddings" should be "three network embedding methods/techniques";

Line 226: for me it is not clear in which settings the methods tend to outperform others;

Line 260: replace "After that" with a synonyme, since you use it too often in the paper;

Line 264: split into train test -> split into train set and test set / split into train and test sets;

Line 270: I guess regression classifier is trained on the learned embedding of graphs in the train set but not on the train set itself;

Line 291: the by far best model -> By far the best model;

Line 303: What is the absolute number of samples? I did not understand what the word "absolute" meant in this context;

*>> We mean the number of nodes with texts rather than text size, corrected in the text.*

Line 315: I would rephrase this sentence or at least not use the word "perform" twice in one sentence;

Line 317: DBLP dataset Table 6 one can see, DBLP dataset one can see in Table 6;

Line 318: for me it is unclear what you mean by stability. You could add some explanation;

*>> We rephrased it, we meant similar quality metrics for different number of training labels.*

Table 6: why did you bold the result in Sent2Vec pretrained/50%?

*>> Corrected, it was a typo.*

Line 337: as far as I understand here begins the results for graph embedding methods, so I would add some paragraph title. It is difficult to follow the paper;

Line 342: that for Cora network data -> that the Cora network. Moreover, the last sentence is hard to follow, so I would rephrase it in full.

Line 356: Aain I would add a paragraph title to separate results for graph embedding methods, text embedding methods and fusion methods.

*>> We added a new structure and consistent paragraph order across the paper for different embedding settings.*

Line 385: the same, add a paragraph title.

Line 393: the same, add a paragraph title, for example, "Result for Link Prediction problem."

Line 398: the sentence "It is quite predictable..." is not clear for me.

Line 399: I would not use the word "though" in the scientific paper;

Line 422: I would use the word "entire" instead of "whole", since it is more formal;

Line 424: add a reference for T-SNE algorithm;

*>> We added reference and explanation.*

Line 426: "In the case of GCN activations of the first convolutional layer" is not fully clear. What are GCN activations? Did you mean activation functions?

Line 429: Again, I would not use "though" in the paper. You can write: "However, there are some exceptions..."

Line 439 and 440: it would be very easy to make a clusterization -> it would be very easy to cluster points.

Line 448: "what is more" is a more informal way of saying, so use formal synonyms such as moreover, furthermore.

The last sentence in the discussion is too long. I would split it into two or three sentences.

## Experimental design

In general, experiments are designed in a good way but sometimes experiments lack some details.

In the Validation section, I would add more details about what your input data is. Moreover, in line 260 you split your data into train and test sets for Node Classification problem and then in your tables you consider different percentages of node labels. So I think you could add in Section 3.5.1 that you consider different ratios of node labels and the same for Link Prediction problem;

*>> We have rewritten the experiment setting section, explaining in details how training and validation were made.*

In line 259 it is stated that the input data is preprocessed. How did you preprocess data? Is this preprocessing step described in Section 3.2 for text data? Or did you preprocess differently? If yes, then it would be better to add details. But then I did not find how you preprocessed the graph-structured data.

>> *We add preprocessing description for text data. No specific preprocessing for graph data was made.*

Also, I did not understand the part "nodes of the network are mapped into latent space using the input information". What kind of input information exactly? Do you mean the input data which you mentioned at the beginning of this sentence?

>> *It was just not a good use of language, we mean applying one of the embedding techniques: textual, structural or fusion. Further, it is used as features in downstream tasks like node classification or link prediction.*

Line 269: What are xu and xvin your case? Add more details;
>> *Corrected.*

## Validity of the findings

The experiments compare the accuracy of different fusion methods which allows the audience to comprehend how the text information could impact the results of node classification and link prediction problems.
The data, which is used in the paper, is well-known benchmark graph-structured data available online.
All conclusions are linked to original research questions and results are well-described.

=================================================================

---

## Round 0.3 · accepted · Accept

When preparing and submit the final files, please update Figure 1 according to the corresponding comment given by the first reviewer.

Reviewer 1 ·

Basic reporting

Except for the problem of Figure 1, the updated version look good now. Please update the figure.

Experimental design

N/A

Validity of the findings

N/A